# Factors Associated with the Lifestyle of Pediatric Healthcare Professionals during the COVID-19 Pandemic

**DOI:** 10.3390/ijerph20032055

**Published:** 2023-01-22

**Authors:** Milena Oliveira Coutinho, Jorge Lopes Cavalcante Neto, Luiz Humberto Rodrigues Souza, Magno Conceição das Mercês, Denise Vasconcelos Fernandes, Carla César Fontes Leite, Daniel Deivson Alves Portella, Tatiane Targino Gomes Draghi, Klaus Araújo Santos, Laura Emmanuela Lima Costa, Lívia Leite da Silva Macêdo, Larissa de Souza Correia, Caroline da Silva Barbosa, Amália Ivine Costa Santana, Lucinete Sena de Oliveira, Mônica Maria do Nascimento, Rodrigo Alves dos Santos Silva, David dos Santos Calheiros, Victor Artur Barros de Mendonça, Ricardo Franklin de Freitas Mussi, Rafaela Gomes dos Santos, Márcio Costa de Souza, Maria Elizângela Ramos Junqueira, Valdinei de Freitas Rodrigues, Ravena Araújo de Oliveira, Raiane Dourado de Souza, Uiliam dos Santos Lima

**Affiliations:** 1Department of Life Sciences, State University of Bahia, Salvador 41180-045, Bahia, Brazil; 2Department of Human Sciences, State University of Bahia, Campus IV, Jacobina 44700-000, Bahia, Brazil; 3Department of Education, State University of Bahia, Guanambi 46430-000, Bahia, Brazil; 4Department of Physical Therapy, Federal University of São Carlos, São Carlos 13565-905, São Paulo, Brazil; 5Liga Álvaro Bahia contra Mortalidade Infantil, Salvador 40050-050, Bahia, Brazil; 6Federal University of Bahia, Salvador 40170-110, Bahia, Brazil; 7UnidomPedro University Center, Salvador 40010-020, Bahia, Brazil; 8Faculdade de Desporto, Universidade do Porto, 4099-002 Porto, Portugal; 9Department of Occupational Therapy, Federal University of Sergipe, Lagarto 49100-000, Sergipe, Brazil; 10Health and Technology Teaching Program, State University of Health Sciences of Alagoas, Maceió 57010-300, Alagoas, Brazil; 11Department of Education, State University of Bahia, Teixeira de Freitas 45992-255, Bahia, Brazil

**Keywords:** occupational health, marital status, income, workday, coronavirus

## Abstract

The objective of this study was to analyze the association between sociodemographic and occupational variables with the lifestyle of health professionals in pediatric units during the COVID-19 pandemic. A cross-sectional study was conducted with 407 health professionals working in four pediatric health units in the State of Bahia, northeast Brazil. The Fantastic Lifestyle Questionnaire and a questionnaire with sociodemographic and employment variables self-applied via Google Forms were used. The final binary logistic regression models revealed significant associations between those who received 3 to 5 minimum wages, without a marital relationship and with more than one employment relationship. Data suggest that these sociodemographic and occupational profiles are more exposed to risky lifestyle behaviors. The findings of this study demonstrated the need for greater attention to the health of health professionals in the pediatric field, as well as the promotion of initiatives that seek to value the work of these professionals and reduce the damage caused to their lifestyle, especially in a pandemic scenario.

## 1. Introduction

A lifestyle is the set of habits and customs adopted by people and communities according to their historical-cultural experiences [1], which is related to but different from the quality of life. The World Health Organization (WHO) characterized the term quality of life as the understanding of people about their position in life, taking into account their lifestyles, goals or lifespan [2]. The main point that distinguishes the terms is the fact that quality of life is regarding people’s perceptions or conceptions about life [3], while lifestyle is regarding people’s behaviors or habits in daily life [4]. The World Health Organization (WHO) [5] also described a healthy lifestyle as a goal of living with good health and less exposure to serious illness or early death. A healthy lifestyle enhances a balance in physical and mental health, including a more positive way to live [5].

A healthy lifestyle has been associated with a lower risk of chronic non-communicable diseases (NCDs), which are responsible for 41 million deaths, that is, 70% of all deaths worldwide and 76% of the causes of death in Brazil [6]. Despite the NCD prevention and control measures adopted in recent years by health systems and services, it is observed that the lifestyle has undergone significant changes, mainly as a result of the COVID-19 pandemic [6,7,8].

Composed of a multivariate construct that encompasses the level of physical activity; consumption of tobacco, alcohol and other drugs; healthy eating; stress management; safe behaviors; and job satisfaction [9,10], lifestyle represents the set of everyday actions that reflect on people’s attitudes and values. Conscious habits and actions are associated with the perception of the lifestyle that the individual brings with them. The components of the lifestyle can change over the years, but this only happens if the person consciously sees some behavior that they should include or exclude, in addition to realizing that they are capable of making the changes they so much want [11,12,13].

In turn, work has always been inserted into people’s daily lives, often having negative consequences on the worker’s life, either by the non-recognition of their labor rights, or by aspects of their health and well-being, because in some cases, the work environment is characterized as unhealthy. With the effects of globalization, work processes have become more demanding and complex. However, regardless of intentions, health professionals in particular have a work environment that is considered rich, stimulating and heterogeneous, but with painful factors, such as mental load, concentration pressure and changes in the patient’s risk conditions [14,15].

Healthcare professionals are exposed to a number of conditions that can interfere with their lifestyle. Most of these health professionals resort to multi-employment (secondary activities and in precarious conditions) to supplement their monthly income; they are more exposed to long working hours, stress and caring for others; and often give up taking care of themselves [8,16].

In addition, the COVID-19 pandemic has impacted public health and the economy with more than 255,324,963 confirmed cases and 5,127,696 deaths from the disease worldwide, and more than 34,793,309 confirmed cases and 687,666 deaths in Brazil until 26 October 2022 [17].

The impacts of the pandemic directly affected the lifestyle of the general population, especially health professionals who, by performing essential activities, not only remained at work, but were also exposed to changes in their routine, behaviors and health, such as greater work overload, long hours and occupational stress, interfering with the goal of a healthy lifestyle for these professionals [18,19,20,21].

Although the lifestyle theme has been researched today, especially considering the population of health workers [22,23], there are still few studies with specific groups, considering the demands and work specificities in the different sectors that make up the healthcare units, such as the field of pediatrics [24,25]. In addition, according to the literature [24,25], professionals working in pediatric units are more vulnerable to psychological distress. The understanding of their lifestyle can be an important starting point to design a profile of their behaviors related to work duties in the pediatric field. This information highlights the lack in the literature and justifies this study. 

Professionals working in the field of pediatrics are exposed to a greater load of stress among other variables, in addition to the exposures inherent to health work [26,27]. We believe that this exposure is strongly linked to the illness and health behaviors of these professionals, which demonstrates the need to know the work and sociodemographic factors that are associated with the lifestyle of these workers, so that it is possible to adopt strategies to protect this target population, especially in a pandemic scenario. In addition, we still do not know if there are different challenges among the pediatric team and other health work categories since the literature has been focused on the overall health workers. Thus, the objective of this study was to analyze the association between sociodemographic and occupational variables with the lifestyle of health professionals in pediatric units during the COVID-19 pandemic. We believe we might build a rationale in the pediatric literature through our results.

## 2. Materials and Methods

This is an exploratory cross-sectional study.

### 2.1. Participants and Study Location

The study sample consisted of 407 health professionals who worked in four pediatric care units, located in the cities of Salvador, Lauro de Freitas and Feira de Santana in the State of Bahia, northeast Brazil. The sample consisted of professionals who were active in the units during the period of data collection, and professionals who were pregnant or on social security leave were excluded from the sample. For ethical reasons, the names of the units were replaced by numbers, as shown in Table 2 (see the results section), to protect the information collected in the units.

Data collection took place during the months of July and August 2021, when health professionals were still working in the COVID-19 pandemic. However, at that time there was an important reduction in morbidity and mortality caused by the disease due to vaccination; it should be noted that health professionals were part of the priority framework for the use of the vaccine.

### 2.2. Sample Calculation

A total of 3204 health professionals were considered eligible to participate in the study according to data from November 2020, made available by the People Management of the participating units. The sample size calculation was performed on the Openepi platform (https://www.openepi.com/SampleSize, accessed on 14 August 2022), considering a 5% alpha and an 80% statistical power, with a 95% confidence interval, with reference to the estimated prevalence of 27% of a sedentary lifestyle among health professionals in the northeast [28] for a 1.2 odds ratio (OR). The prevalence of a sedentary lifestyle was considered as a reference because physical activity represents one of the main domains of a lifestyle, and due to the absence of specific references reporting the prevalence of an inappropriate lifestyle considering all domains. The minimum estimated sample was 333 health professionals. Taking into account possible sample losses, 20% more was added to the sample, which totals 400 health professionals as the minimum sample expected for the study. Finally, a total of 407 professionals made up the final sample of the study.

### 2.3. Variables and Instruments

#### 2.3.1. Lifestyle

The “Fantastic Lifestyle” questionnaire was used to assess the workers’ lifestyle. The instrument showed good psychometric qualities in the validation study with young Brazilian adults, confirmed by a Cronbach’s alpha of 0.69 and 0.60 considering the grouping of domains. The construct validity attested to by the concordance rate (CR) considering the four categories of the instrument (Regular, Good, Very Good and Excellent) was 75%, and the concordance rate for nominal scale (Kappa) was 0.70, considering the two applications of the instrument, which found a classification ration in three groups of 80.7%. In addition, the questionnaire showed high intraclass reproducibility (R = 0.92), attesting to the instrument’s ability to assess lifestyle in adults [29].

This instrument was duly validated in Brazil [29], was self-applied and had twenty-five questions divided into nine domains or variables: (1) family and friends; (2) physical activity; (3) nutrition; (4) tobacco and drugs; (5) alcohol; (6) sleep, seat belt, stress and safe sex; (7) type of behavior; (8) introspection; (9) work. The questions were coded by points on a scale ranging from 0 (lowest score) to 4 (highest score), and some items had only two response options, which were 0 or 4. Originally, the sum of all points results in a score which classifies the lifestyle of individuals into five categories, as follows: “Excellent” (85 to 100 points), “Very good” (70 to 84 points), “Good” (55 to 69 points), “Fair” (35 to 54 points) and “Needs improvement” (0 to 34 points). However, for the purposes of analyzing the associations in this study, we dichotomized the final score from the median value, classifying participants into two groups: higher lifestyle scores (≥74 points) and lower lifestyle scores (<74 points).

#### 2.3.2. Sociodemographic Characteristics and Labor Issues

A specific questionnaire was developed by the researchers to assess these characteristics, consisting of 39 closed, multiple-choice, self-administered questions.

The sociodemographic variables eligible for the study were:-Sex (female or male);-Age (dichotomized into the age group of 18–34 years and 35–64 years, based on the median age distribution);-Residence (countryside or urban);-Self-reported race/color (white, brown, black, yellow, indigenous, without declaration);-Performs domestic work activities (no or yes);-Frequency of domestic activity (none, 1 to 3 days a week, daily);-Family income (1 minimum wage, 1 to 2 minimum wages, 3 and ≥ 5 minimum wages);-Number of people residing in the household (1, 2 to 3, 4 to ≥ 8);-Type of ownership of the house (own, rented, other);-Marital status (single, widowed, divorced—no marital relationship; married, stable union—with marital relationship);-Number of children (none, from 1 to 2, from 3 to 4, ≥ 5);

Eligible labor variables for the study were:
-Unit (1, 2, 3, 4);-Training to perform the current role (yes or no);-Work sector (Intensive Care Unit—ICU, Inpatient Clinics, Transfusion Agency, Infection Control Service—SCIH, Outpatient, Administrative, Operation Assistance, Management, Multi-Team, Services, Surgical or Obstetric Center, Oncology, Intermediate Care Unit);-Assistance group: (i) Indirect healthcare (administrators, administrative assistants, stretcher workers, janitors, cleaning assistants, nutrition professionals, managers, receptionists); (ii) direct healthcare (nurses, doctors, nursing technicians, physiotherapists, social workers, psychologists, speech therapists, physical educators);-Working time in the current contract (in years);-Weekly working hours in the current contract (in hours);-Works at another location (no or yes);-Commuting to work: (i) Active (on foot or by bicycle); (ii) Passive (by bus, car, motorcycle or van);-Commuting back from work: (i) Active (on foot or by bicycle); (ii) Passive (by bus, car, motorcycle or van);-Night shift (no or yes);-Type of employment relationship (statutory, contract, service provider, other).

### 2.4. Procedures

The target audience received, via email, chat application and through the units’ formal means of communication, the access link to the questionnaires and the Informed Consent Form—ICF to participate in the survey through the Google Forms platform during the months of July and August 2021. To verify the consistency of the questions and access to the instruments and ICF via Google Forms, the researchers carried out a pilot study, which can attest to the feasibility of the instruments and certify the training of the evaluators, who were experienced researchers in this area of study.

### 2.5. Statistical Analysis

Bivariate association analyses were performed with the chi-square test and odds ratio (OR) calculation with a 95% confidence interval, with lifestyle dichotomized (higher lifestyle scores ↑ and lower lifestyle scores ↓) as the dependent variable, and sociodemographic and work factors as the independent variables of the study. The significance level adopted was 5%. In addition, independent variables with values of *p* ≤ 0.20 in the bivariate analysis were included in the proposed binary logistic regression models to explain the predictive capacity of the independent variables that remained associated with lifestyle in the final logistic regression models presented in this study. Considering the number of independent variables and the specificities of these eligible variables, two logistic regression models were calculated: the first considering exclusively the sociodemographic variables with *p* ≤ 0.20, and the second considering exclusively the labor variables with *p* ≤ 0.20. To perform the statistical calculations, SPSS version 20.0 for Windows was used.

## 3. Results

The final sample of the study consisted of 407 workers from the researched pediatric units, which correspond to 85.01% from Unit 1, 6.39% from Unit 2, 5.41% from Unit 3 and 3.19% from Unit 4. Most participants were female (81.8%), aged between 18 and 34 years (51.6%) and of mixed race (56.5%).

Table 1 presents the analysis of associations between sociodemographic and lifestyle variables of the health professionals. The variables that were significantly associated with the lifestyle of health professionals were as follows: housework (OR = 3.49; CI: 95% = 1.51–8.0; *p* < 0.01), frequency of domestic activity (*p* < 0.01), family income (*p* = 0.02) and marital status (OR = 1.75; CI: 95% = 1.18–2.61; *p* < 0.01).

Table 2 presents the analysis of the associations between work variables and health professionals’ lifestyle. There was a significant association between working at another location (OR = 0.63; CI: 95% = 0.42–0.94; *p* = 0.02) and the health professionals’ lifestyle.

Table 3 shows the final binary logistic regression model of the associations between sociodemographic and lifestyle variables of the health professionals. Among the variables included in the initial model, family income and marital status remained associated with the health professionals’ lifestyle. We found an R^2^ of 0.150 in the final logistic regression model, which explains 15% of the lifestyle of the participating workers.

Table 4 shows the final binary logistic regression model of the associations between work variables and the health professionals’ lifestyle. The variable “works elsewhere” remained associated with the lifestyle of health professionals. The analysis showed an R^2^ of 0.124 in the final logistic regression model, which explains approximately 12.4% of the lifestyle of the participating workers.

## 4. Discussion

The present study aimed to analyze the association between sociodemographic and occupational variables with the lifestyle of health professionals in pediatric units. The results demonstrate that the lifestyle of the participants was considered very good, where most participants (53.8%) achieved scores ≥ 74 points, that is, higher scores for lifestyle. This finding demonstrates that, in general, the lifestyle of these workers can be considered adequate through the lifestyle construct of the Fantastic Questionnaire.

This result is similar to the data found by Rocha et al. [30] in health professionals working in different sectors, where 92.8% of these professionals had a lifestyle considered adequate. Other studies [31,32] corroborate our findings, strengthening the relevance that, despite the occupational, lifestyle and socioeconomic risks to which they are exposed, especially in the face of the pandemic state, most health professionals manage to achieve a health-promoting lifestyle [33,34]. A factor to be observed is that, in order to participate in the study, the worker needed to be in full exercise of their work functions: a fact that demonstrates the possibility of the occurrence of “healthy worker bias” and “sampling bias”, where the participation of these workers may have been more represented by these being healthy and with a greater chance of presenting better lifestyles [35].

However, some studies, which used other scales to verify lifestyle, have shown an unhealthy lifestyle in health professionals [9,22,30,31,32,33,34], also demonstrating that an expressive number adopt an unhealthy lifestyle, with inadequate food; tobacco consumption; soft drinks, teas and coffee; few hours of sleep; exhausting working hours; sedentary lifestyle, among other behaviors that predict chronic diseases. Recent studies [36,37] state that, during the COVID-19 pandemic, being an essential worker was a risk factor for a lifestyle considered poor, in addition to reporting that the pandemic had an impact on the use of substances, sleeping and eating habits of these professionals, leading to lower scores in the assessment of lifestyle.

It should be noted that health professionals build their life habits during professional training, as well as in their professional practice and daily routines. Therefore, although the fact that the lifestyles of most of these workers have achieved higher scores in the Fantastic Questionnaire is promising, even in a pandemic scenario, this population lacks interventions that promote changes in lifestyle. These interventions would directly impact the living and health conditions of these professionals, as well as the work situations in which they are inserted [36,37,38].

Significant associations between independent sociodemographic variables (family income and marital status) and labor (more than one employment relationship) and lifestyle remained in the final logistic regression models, and an in-depth discussion is necessary to better understand the lifestyle-related behaviors of this population and their potential implications.

### 4.1. Sociodemographic Factors Associated with Lifestyle

In the binary logistic regression analysis, health professionals with a family income in the range of 3 to 5 minimum wages and without a marital relationship had significantly lower lifestyle scores than the other family income groups. This demonstrates that, despite achieving good numbers related to income, in which they are expected to make better health choices, these professionals give up care for their health; probably giving up their rest; increasing their working hours; reducing their sleep hours; making unhealthy choices related to food, physical activity and substance use such as tobacco and alcohol; in addition to facing greater exposure to stress.

Observing the importance of lifestyle in people’s health conditions, Santos, Jacinto and Tejada [39] and Bomfim et al. [31] argued that a higher income allows for greater access to consumer goods, services, health, education and housing, as income affects health in the sense of more access and opportunities. A study carried out with university students [31] showed that the higher the income, the better their lifestyle, while research carried out with Dominican adults [40] showed that people with lower incomes tend to have healthier lifestyle behaviors and lower chances of morbidity.

Health professionals commonly opt for many contracts and longer working hours in the search for better remuneration [41]. However, access to higher incomes is not always linked to a lifestyle considered adequate.

Related to marital status, health professionals without a marital relationship, that is, those who are single or divorced, were 1.75 times more likely (*p* < 0.01) to have low lifestyle scores than their peers in a marital relationship. This finding demonstrates that professionals without a marital relationship seem to adopt behaviors that involve health risks more frequently.

Studies carried out with the adult population [31,42] showed that single people were more likely to have an inadequate lifestyle because they had accumulated risk factors related to drinking and smoking, and affective relationships influence these behaviors. Regarding the lifestyle and marital status of health professionals, Fernandes et al. [32] observed that most professionals who reported having a lifestyle considered excellent were married, while singles understood their lifestyle as regular.

Having a stable marital relationship is linked to the cultivation of well-being and healthier choices and behaviors [32]. Therefore, maintaining affective relationships, which involve some type of marital relationship, seems to enhance a lifestyle with higher scores, highlighting the importance of health-promoting strategies that achieve higher lifestyle scores also among health professionals without a marital relationship, such as singles or divorcees.

### 4.2. Labor Factor Associated with Lifestyle

With regard to work and lifestyle, working in another location, that is, having more than one employment relationship, was significantly associated with low scores in the lifestyle of health professionals (*p* = 0.02). Exposure to more than one bond and long working hours seem to have an impact on the availability of health professionals for leisure time, regular physical activity, adoption of healthy eating, opting for meals outside the home and practices, reduction in sleep hours and greater stress conditions, which directly influence lifestyle, leading to low scores on the Fantastic Lifestyle Questionnaire.

Professionals who work with health in hospitals commonly choose more than one employment relationship, because the workday in these units is usually in 12 h shifts, alternating with 36 h off, which should be used for rest and personal care. However, the adoption of long hours, accumulation of employment relationships and the search for better financial remuneration have been common, and exert impacts strongly associated with low lifestyle scores [43].

Previous studies with health professionals showed that most of the professionals evaluated usually accumulate several employment relationships and long working hours [43]. However, it is worth noting that the COVID-19 pandemic has had negative impacts on professional working hours and overload, especially having a greater impact on professionals with double working hours due to excessive exposure [44,45], increasing the risk of illness and Burnout Syndrome. These findings indicate that policies for professional financial recognition are necessary, as well as strategies to promote and protect the health of these exposed professionals. Considering the specificity of pediatric health work, it became difficult to compare these results with other studies, as no studies were found that used the Fantastic Lifestyle Questionnaire to assess this group of professionals. Therefore, this study highlights sociodemographic and work factors associated with the lifestyle of health professionals working in pediatrics during the pandemic in the literature.

As limitations of the study, we can highlight the fact that the research was carried out only with workers active in their work functions, which may have led to the exclusion of workers from the sample who may have been on leave due to health issues, possibly work-related. Specific analyses are relevant to better explain the associations of sociodemographic and labor variables and their relationship with lifestyle, considering the long-term impacts of this outcome, as well as the impacts related to the COVID-19 pandemic on this group of workers. Finally, the study was only conducted in a limited number of hospitals in Bahia, Brazil, and caution should be taken to generalize the results.

## 5. Conclusions

The final binary logistic regression models revealed significant associations between family income, marital status, other employment and the lifestyle of pediatric unit workers. The data found indicate that lower lifestyle scores were more evident among those who earned from 3 to 5 minimum wages, without a marital relationship and with more than one employment relationship. It is suggested that these sociodemographic and occupational profiles are more exposed to risk behaviors related to lifestyle.

The findings of this study demonstrate the need for greater attention to the health of health professionals in the pediatric field, as well as the promotion of initiatives that seek to value the work of these professionals and reduce the damage caused to their lifestyle. Since the pandemic is under control, this study is relevant to build further rationale in relation to sociodemographic factors, work factors and healthy lifestyle.

## Figures and Tables

**Table 1 ijerph-20-02055-t001:** Bivariate analysis of the association between lifestyle dichotomized into minor (↓) and major (↑) scores and sociodemographic variables of health professionals in pediatric units.

Variables	Lifestyle		*p*-Value	OR (CI: 95%)
	↓ Scores *n* (%)	↑ Scores *n* (%)		
Sex			0.13	0.68 (0.41–1.12)
Female	148 (78.7)	185 (84.5)		
Male	40 (21.3)	34 (15.5)		
Age group			0.42	1.17 (0.79–1.73)
18 to 34	101 (53.7)	109 (49.8)		
35 to 64	87 (46.3)	110 (50.2)		
Residence			0.78	1.17 (0.37–3.69)
Countryside	6 (3.2)	6 (2.7)		
Urban	182 (96.8)	213 (97. 3)		
Race/Color				___________
White	26 (13.8)	34 (15.5)		
Brown	109 (58.0)	121 (55.3)		
Black	44 (23.4)	61 (27.9)		
Yellow	2 (1.1)	2 (0.9)		
Indigenous	1 (0.5)	1 (0.5)		
without/declaration	6 (3.2)	0 (0.0)		
Housework No Yes	22 (11.7) 166 (88.3)	8 (3.7) 211 (96.3)	<0.01	3.49 (1.51–8.05)
Frequency of domestic activity None 1–3 days Daily	22 (11.7) 79 (42.0) 87 (46.3)	7 (3.2) 91 (41.6) 121 (55.3)	<0.01	___________
Family income (in minimum wages) Above 5 Between 3 to 5 1 to 2 1	22 (11.7) 76 (40.4) 62 (33.0) 28 (14.9)	42 (19.2) 61 (27.9) 76 (34.7) 40 (18.3)	0.02	___________
Type of domicile Own Rent Other	132 (70.2) 43 (22.9) 13 (6.9)	169 (77.2) 41 (18.7) 9 (4.1)	0.22	___________
Number of residents in the household <4 people ≥4 people	126 (67.0) 62 (33.0)	153 (69.9) 66 (30.1)	0.53	0.87 (0.57–1.33)
Marital status No marital relationship With marital relationship	117 (62.2) 71 (37.8)	106 (48.4) 113 (51.6)	<0.01	1.75 (1.18–2.61)
Number of children None 1 to 2 3 to 4 ≥5	93 (49.5) 82 (43.6) 11 (5.9) 2 (1.1)	100 (45.7) 110 (50.2) 9 (4.1) 0 (0.0)	0.17	___________
Education level Middle level Technical level University graduate Specialization Master’s degree	32 (17.0) 32 (17.0) 42 (22.3) 77 (41.0) 5 (2.7)	33 (15.1) 53 (24.2) 40 (18.3) 83 (37.9) 10 (4.6)	0.30	___________

OR: odds ratio; CI: confidence interval.

**Table 2 ijerph-20-02055-t002:** Bivariate analysis of the association between lifestyle dichotomized into minor (↓) and major (↑) scores and work variables of health professionals in pediatric units.

Variables	Lifestyle		*p*-Value	OR (CI: 95%)
	↓ Scores *n* (%)	↑ Scores *n* (%)		
Unit 1 2 3 4	16 (8.5) 154 (81.9) 11 (5.9) 7 (3.7)	10 (6.4) 192 (87.7) 11 (5.0) 6 (2.7)	0.35	___________
Qualification to perform the role Yes No	174 (92.6) 14 (7.4)	211 (96.3) 8 (3.7)	0.09	0.47 (0.19–1.14)
Sector			0.14	___________
ICU	33 (17.6)	26 (11.9)		
Inpatient Clinics	31 (16.5)	29 (13.2)		
Transfusion Agency	1 (0.5)	1 (0.5)		
SICH	2 (1.1)	0 (0.0)		
Outpatient	4 (2.1)	5 (2.3)		
Administrative	20 (10.6)	25 (11.4)		
Assistance Operation	4 (2.1)	14 (6.4)		
Management	12 (6.4)	9 (4.1)		
Emergency	2 (1.1)	5 (2.3)		
Multi-Team	16 (8.5)	33 (15.1)		
Services	36 (19.1)	35 (16.0)		
Surgical or Obstetric Center	12 (6.4)	16 (7.3)		
Oncology	6 (3.2)	12 (5.5)		
Intermediate Care Unit	9 (4.8)	9 (4.1)		
Health assistance Indirect Assistance Direct Assistance	62 (33.0) 126 (67.0)	65 (29.7) 154 (70.3)	0.47	1.16 (0.76–1.77)
Works elsewhere No Yes	106 (56.4) 82 (43.6)	147 (67.1) 72 (32.9)	0.02	0.63 (0.42–0.94)
Night duty No Yes	83 (44.1) 105 (55.9)	115 (52.5) 104 (47.5)	0.09	0.71 (0.48–1.05)
Type of employment relationship Statutory Signed Portfolio Service Provider Other	0 (0.0) 183 (97.3) 5 (2.7) 0 (0.0)	2 (0.9) 207 (94.5) 6 (2.7) 4 (1.8)	0.05	___________
Commute to work Activ ePassive	4 (2.1) 184 (97.9)	6 (2.7) 213 (97.3)	0.70	0.77 (0.21–2.77)
Commute back from work Active Passive	3 (1.6) 185 (98.4)	5 (2.3) 214 (97.7)	0.62	0.69 (0.16–2.94)

OR: odds ratio; CI: confidence interval.

**Table 3 ijerph-20-02055-t003:** Final model of binary logistic regression of sociodemographic factors associated with the lifestyle of workers participating in the study.

Likelihood of Associated Factors	B	Exp (B)	*p*-Value	Constant	−2 log
Family income	−0.945	0.389	<0.01	−42.060	513.457
Marital status	−0.664	0.515	<0.01		

**Table 4 ijerph-20-02055-t004:** Binary logistic regression analysis of the association between lifestyle and work variables of health workers participating in the study.

Likelihood of Associated Factors	B	Exp (B)	*p*-Value	Constant	−2 log
Works elsewhere	0.670	1.953	<0.01	20.032	522.317

## Data Availability

Not applicable.

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
