# Peer review of "Factors Associated with the Lifestyle of Pediatric Healthcare Professionals during the COVID-19 Pandemic"

_ijerph, 2023, doi:10.3390/ijerph20032055_

Round 1

Reviewer 1 Report (New Reviewer)

Since there are paragraphs written in red, I assume that this form of the article is an improved one.
I recommend that the abstract has the same form as the article: introduction, methods, results, conclusions, marking them in the abstract.

Also, it seems to be too many authors..., in contribution area, only 5 did the investigation.

Author Response

#Reviewer 1

Since there are paragraphs written in red, I assume that this form of the article is an improved one.

Dear reviewer, yes. The article was improved according to previous review report from the Journal.

I recommend that the abstract has the same form as the article: introduction, methods, results, conclusions, marking them in the abstract.

We appreciate your suggestion, but the structure is following the Journal’s format.

Also, it seems to be too many authors..., in contribution area, only 5 did the investigation.

This was an observation already pointed out by the editors and we proper explained each author contribution and the relevance of each one in the article. Please, see the explanation below:

This study originated from a Master’s thesis of MOC, who was supervised by JLCN. The professors LHRS and MCM were on the board in this defense and provided great insightful comments on the work. DVS and CCFL are both Master’s students and helped MOC in her data collection and analysis. After the thesis defense the statistical analysis used in the manuscript was checked by DDAP, TTGD, RFFM and MERJ, who made great improvements. KAS, LELC, RASS, DSC, RGS and MCS provided a huge literature review to build along with the other authors a strong rationale of this study. LLSM, LSC, CSB and AICS worked at the hospitals and supported all the recruitment of the participants and data collection with MOC. Finally the students LSO, MMN, VABM, VFR, RAO, RDS and USL are member of the lab and helped with the data collection, data tabulation, statistical software setting and resources. They double-checked the data entry in the systems before send it to the other members of the team to analyse the data.

Reviewer 2 Report (New Reviewer)

I have enjoyed reading this article. The paper is clearly, concisely, and logically written. Results are clearly described. The discussion, limitations and conclusion are well-presented. However, I have some points to make it better:

Dear Authors,

1. Authors mentioned “l few studies with specific groups, considering the demands and work specificities in the different sectors…. “ in the line 100. Please mention some of these researches in this part of the paper to better identify the research gaps.

2. Please remove some information at the first paragraph of Materials and Methods section. This information is redundant and mentioned in the Institutional Review Board Statement section at the end of the paper.

3. In the limitations of the research, it should be noted that this study was conducted in a limited number of hospitals in Bahia, Brazil, and more caution should be taken to generalize the results.

4. Some sentences can be expressed better in terms of the English language proficiency such as lines 381-384. Please proofread all sentences in the text.

Author Response

#Reviewer 2

I have enjoyed reading this article. The paper is clearly, concisely, and logically written. Results are clearly described. The discussion, limitations and conclusion are well-presented. However, I have some points to make it better:

Thank you for the compliments and for the points to improve our study.

Dear Authors,

  1. Authors mentioned “l few studies with specific groups, considering the demands and work specificities in the different sectors…. “ in the line 100. Please mention some of these researches in this part of the paper to better identify the research gaps.

We mentioned the studies as recommended by the reviewer:

Although the lifestyle theme has been researched today, especially considering the population of health workers [22, 23], there are still few studies with specific groups, considering the demands and work specificities in the different sectors that make up the health care units, such as the field of pediatrics [24, 25]. (Introduction section, Page 3, Line 105).

  1. Please remove some information at the first paragraph of Materials and Methods section. This information is redundant and mentioned in the Institutional Review Board Statement section at the end of the paper.

We removed the duplicate information at the first paragraph of the Materials and Methods section following the reviewer suggestion. Only the information of the study design is there now as follows:

This is an exploratory cross-sectional study. (Materials and Methods section, page 3, Line 124).

  1. In the limitations of the research, it should be noted that this study was conducted in a limited number of hospitals in Bahia, Brazil, and more caution should be taken to generalize the results.

Thank you for this observation. We included the information as a limitation of the study as follows:

Finally, the study was only conducted in a limited number of hospitals in Bahia, Brazil and caution should be taken to generalize the results. (Discussion section, Page 13, Lines 394-395).

  1. Some sentences can be expressed better in terms of the English language proficiency such as lines 381-384. Please proofread all sentences in the text.

We changed the terms and adapting the sentence for better English, as follows:

Therefore, this study highlights sociodemographic and work factors associated with the lifestyle of health professionals working in pediatrics during the pandemic in the literature. (Discussion section, Page 13, Lines 385-387).

Reviewer 3 Report (New Reviewer)

Notes formally:

- Why are some parts of the paper written in red?

- In tab. 1 – 4 source is missing

- There are abbreviations in the paper that are not explained

Content notes:

- The objective is defined in the abstract: "The objective of this study was to identify the sociodemographic and occupational variables capable of predicting the lifestyle of health professionals in pediatric units during the 40 COVID-19 pandemic".

     (i) Variables in relation to another variable, in this case lifestyle, are referred to as predictors. Predictors explore multiple authors (doi: 10.1016/j.ypmed.2020.106061; DOI: 10.3390/ijerph19106185)

     (ii) "Predicting the lifestyle" – for what period? 1 year? 10 years?.

Prediction is about predicting. There is no mention of lifestyle prediction in the paper. It follows that the goal is not in line with the content of the paper. I recommend reworking it.

- The key point of the paper is lifestyle. I recommend the authors to briefly describe lifestyle (10.1080/02614369300390231; https://doi.org/10.3141/2495-08), including healthy lifestyle (https://doi.org/10.1161/CIRCULATIONAHA.117.032047) defined by WHO (https:// apps.who.int/iris/handle/10665/108180)

- Paper concerns one part of the quality of life of pediatric professionals, which is the Quality of work life or Quality of work - life balance (https://doi.org/10.1177/0971685820939). I recommend describing them in a few lines.

- Line 167 et seq.: "The result is a division into 5 groups". Why do the groups have unequal value ranges?

Author Response

#Reviewer 3

Notes formally:

- Why are some parts of the paper written in red?

Dear reviewer, because the paper was improved according to previous review report from the Journal.

- In tab. 1 – 4 source is missing

Dear reviewer, according to the Journal’s norms this is not applicable.

- There are abbreviations in the paper that are not explained

We checked over the manuscript and the missing explanation of the abbreviation was made.

Content notes:

- The objective is defined in the abstract: "The objective of this study was to identify the sociodemographic and occupational variables capable of predicting the lifestyle of health professionals in pediatric units during the 40 COVID-19 pandemic".   (i) Variables in relation to another variable, in this case lifestyle, are referred to as predictors. Predictors explore multiple authors (doi: 10.1016/j.ypmed.2020.106061; DOI: 10.3390/ijerph19106185) (ii) "Predicting the lifestyle" – for what period? 1 year? 10 years?. Prediction is about predicting. There is no mention of lifestyle prediction in the paper. It follows that the goal is not in line with the content of the paper. I recommend reworking it.

We appreciate the reviewer for these huge literature examples to help us improving our rationale, analysis and ideas. We agreed with the fact prediction is about predicting and in our paper we were able to analyse association between variables, particularly. According to your comments, we changed the objective of the study in order to fit a proper way we explored the analysis and discussion, as follows:

The objective of this study was to analyze the association between sociodemographic and occupational variables with the lifestyle of health professionals in pediatric units during the COVID-19 pandemic.

- The key point of the paper is lifestyle. I recommend the authors to briefly describe lifestyle (10.1080/02614369300390231; https://doi.org/10.3141/2495-08), including healthy lifestyle (https://doi.org/10.1161/CIRCULATIONAHA.117.032047) defined by WHO (https:// apps.who.int/iris/handle/10665/108180)

We described accordlying, as follows:

The World Health Organization (WHO) [5] also described a healthy lifestyle as a goal of living with healthy and less exposure to serious illness or early death. Health lifestyle enhances a balance in physical and mental health, including a more positive way to live [5]. (Introduction section, Page 2, Lines 62-65).

- Paper concerns one part of the quality of life of pediatric professionals, which is the Quality of work life or Quality of work - life balance (https://doi.org/10.1177/0971685820939). I recommend describing them in a few lines.

Dear reviewer, thank you for this insightful observation. Despite we can find relation between quality of life and lifestyle we would like to put each construct apart. This was a previous concern from another reviewer in the first round and we rewrote the introduction to follow this. Our concern describing quality of life of pediatric professionals is raising another point for another construct and missing the focus of this study. We are sorry for that and believe the reviewer can understand it.

- Line 167 et seq.: "The result is a division into 5 groups". Why do the groups have unequal value ranges?

Because there are domains with more items and domains with less items.

More details of the fantastic questionnaire and its scores can be checked as follows:

https://cpb-ca-c1.wpmucdn.com/myriverside.sd43.bc.ca/dist/6/45/files/2014/01/Fantastic-Lifestyle-Checklist-Fillable-1smptgc.pdf

https://www.scielo.br/j/abc/a/hZygGvfLfbMRL44bjzjCPKh/?lang=en&format=pdf

Reviewer 4 Report (New Reviewer)

Dear Authors, thanks for the opportunity to review this work.

It is noted the effort made by the authors, although, in the introduction, discussion, and conclusion, I do not find a  scientific reason for this study. What is the lack in the literature that justifies the study? The findings are not novel compared to a time without COVID-19. The authors must present the construct validity analysis of the instrument used. Results must follow APA norms. Since the pandemic is almost ending, the authors should point out the importance of this publication for research worldwide.

Author Response

#Reviewer 4

Dear Authors, thanks for the opportunity to review this work.

We appreciate the reviewer for the compliment.

It is noted the effort made by the authors, although, in the introduction, discussion, and conclusion, I do not find a  scientific reason for this study. What is the lack in the literature that justifies the study? The findings are not novel compared to a time without COVID-19.

Dear reviewer, we understand your concern. We have improved our rationale according to the previous review reports and the lack in the literature is now included in our introduction as follows:

In addition, according to the literature [24, 25] professionals working in pediatric units are more vulnerable to psychological distress. The, understanding their lifestyle can be an important starting point to design a profile of their behaviors related to work duties in pediatric field. This information highlights the lack in the literature and justifies this study. (Introduction section, Page 3, Lines 105-109).

The authors must present the construct validity analysis of the instrument used.

Dear reviewer, we already presented those data in the article. Un update was made to be clear about the construct validity analysis used for the validation process of the instrument, as follows:

The construct validity attested by the concordance rate (CR) considering the four categories of the instrument (Regular, Good, Very Good and Excellent) was 75% and the concordance rate for nominal scale (Kappa) was 0.70, considering the two applications of the instrument, which found a classification ration in three groups of 80.7%. In addition, the questionnaire showed high intraclass reproducibility (R = 0.92), attesting to the instrument's ability to assess lifestyle in adults [29]. (Material and Methods section, Page 4, Lines 160-165).

Results must follow APA norms.

Dear reviewer, we do not understand your comment since the Journal does not follow APA norms.

Since the pandemic is almost ending, the authors should point out the importance of this publication for research worldwide.

Since the pandemic is controlled, this study is relevant to build further rationale in relation to sociodemographic, work factors and health lifestyle. (Conclusion section, page 13, lines 406-407).

This manuscript is a resubmission of an earlier submission. The following is a list of the peer review reports and author responses from that submission.

Round 1

Reviewer 1 Report

Dear authors,

I had the opportunity to make the review for your article.

Reading your introduction I am quite perplexed about the lifestyle concept and construct and the reference sample. 

From the description of the construct you provide, it seems that you refer to the quality of life of the individuals and not to their specific lifestyles. Similarly, the reference to this sub-category of health workers is not very clear. Covid-19 has certainly challenged all health professionals, but why did you want to focus on paediatric ones? What different challenges and what scientific significance do they have?

In relation to the Fantastic Lifestyle instrument, the alpha seems a bit low to me. Can you add any measures of reliability? AVE and CR for example?

I don't understand the logic of dividing the high and low points of the lifestyle score. Taking into account the tool and the indications you provide saying that scores below 74 are all indicative of a low lifestyle seems far fetched to me.

Why do you use a p value of .20? I fail to see the point of the results given this significance index.

I suggest you take a closer look at the study model considered.

Author Response

Manuscript ID: ijerph-2025232

Title: Factors associated with the lifestyle of pediatric healthcare
professionals during the COVID-19 pandemic

Dear Editors,           

We thank the reviewers for their very helpful and insightful comments and suggestions. We have considered these very carefully and revised our paper accordingly, which has enhanced its quality and impact. 

Please, find below (in red) the responses to the points raised by the reviewers.

#Reviewer 1

Dear authors,

I had the opportunity to make the review for your article.

Reading your introduction I am quite perplexed about the lifestyle concept and construct and the reference sample. 

From the description of the construct you provide, it seems that you refer to the quality of life of the individuals and not to their specific lifestyles.

Dear reviewer, thank you for your precise observation. Lifestyle and quality of life are related terms, but with different constructions and rationales. In order to elucidate it, the beginning of our introduction was reformulated as follows:

Lifestyle is the set of habits and customs adopted by people and communities according to their historical-cultural experiences [1], which is related to it but different from the quality of life. The World Health Organization (WHO) characterized the term quality of life as the understanding of people about their position in life, taking into account their lifestyles, goals or lifespan[2]. The main point that distinguishes the terms is the fact that quality of life is regarding people’s perceptions or conceptions about life[3], while lifestyle is regarding people’s behaviors or habits in daily life[4].  (Introduction Section, Page 2, Lines 56-62).

Similarly, the reference to this sub-category of health workers is not very clear. Covid-19 has certainly challenged all health professionals, but why did you want to focus on paediatric ones? What different challenges and what scientific significance do they have?

We appreciate the reviewer for this insightful comment. We improved the introduction as follows:

In addition, we still do not know if there are different challenges among the pediatric team and other health work categories since the literature has got focused on the overall health workers.

Thus, the objective of this study was to identify the sociodemographic and occupational variables capable of predicting the lifestyle of health professionals in pediatric units during the COVID-19 pandemic. We believe we might build a rationale in the pediatric literature through our results. (Introduction Section, Page 3, Lines 108-114).

In relation to the Fantastic Lifestyle instrument, the alpha seems a bit low to me. Can you add any measures of reliability? AVE and CR for example?

We agree with the reviewer about the low value of the alpha. However, it is still an acceptable value of alpha for this type of instrument, which was also pointed out by the authors of the validation study.

Rodriguez Añez, Ciro Romélio, Reis, Rodrigo Siqueira e Petroski, Edio Luiz. Versão brasileira do questionário "estilo de vida fantástico": tradução e validação para adultos jovens. Arquivos Brasileiros de Cardiologia [online]. 2008, v. 91, n. 2 [Acessado 9 Dezembro 2022], pp. 102-109. Disponível em: <https://doi.org/10.1590/S0066-782X2008001400006>. Epub 08 Ago 2008. ISSN 1678-4170. https://doi.org/10.1590/S0066-782X2008001400006.

Following the reviewer recommendation, we added this information in the manuscript as follows:

The concordance rate (CR) considering the four categories of the instrument (Regular, Good, Very Good and Excellent) was 75% and the concordance rate (Kappa) was 0.70. (Methods section, Page 4, Lines 157-159).

I don't understand the logic of dividing the high and low points of the lifestyle score. Taking into account the tool and the indications you provide saying that scores below 74 are all indicative of a low lifestyle seems far fetched to me.

We understand the reviewer’s concerns about dividing the scores into two categories, but we needed turning our dependent continuous variable (lifestyle) into categorical variable (dichotomous) to perform the associations’ analyses in SPSS. However, aiming to find the best way to categorize our dependent variable we used the medians as reference because there is no dichotomous cut-offs for classifying Lifestyle from the Fantastic Lifestyle.

Also, medians are widely used option, despite the costs (DeCoster, Gallucci, & Iselin, (2011). Best practices for using median splits, artificial categorization, and their continuous alternatives. Journal of Experimental Psychopathology, 2(2), 197–209. https://doi.org/10.5127/jep.008310).

Additionally, there is support in the literature from published previous study (see below) to adopt categorized classification (two groups) for lifestyle as a reference, likewise our study.

Cristina de Oliveira N, Alfieri FM, Lima ARS, Portes LA. Lifestyle and Pain in Women With Knee Osteoarthritis. Am J Lifestyle Med. 2017 Jul 25;13(6):606-610. doi: 10.1177/1559827617722112. PMID: 31662727; PMCID: PMC6796231.

We considered the median the best option because it distributed the sample proportionally and allowed the associations’ analyses (DeCoster, Gallucci, & Iselin, 2011). Thus, those participants who scored below 74 were classified as having “lower lifestyle scores↓,” while those who scored equal or above 74 were classified as having “higher lifestyle scores↑”.

Why do you use a p value of .20? I fail to see the point of the results given this significance index.

Because this value is the usual value adopted in the literature to include potential variables from the bivariate analyses to the binary logistic regression models to explain the predictive capacity of the independent variables that remained associated with lifestyle in the final logistic regression models presented in this study.

This explanation is justified in the literature as follows:

Ranganathan P, Pramesh CS, Aggarwal R. Common pitfalls in statistical analysis: Logistic regression. Perspect Clin Res. 2017 Jul-Sep;8(3):148-151. doi: 10.4103/picr.PICR_87_17. PMID: 28828311; PMCID: PMC5543767.

Cavalcante, Jorge Lopes, Sato, Tatiana de Oliveira and Tudella, Eloisa. Socio-demographic factors influences on guardians’ perception of Developmental Coordination Disorder among Brazilian schoolchildren. Motriz: Revista de Educação Física [online]. 2018, v. 24, n. 02 [Accessed 9 December 2022], e101810. Available from: <https://doi.org/10.1590/S1980-6574201800020002>. Epub 17 May 2018. ISSN 1980-6574. https://doi.org/10.1590/S1980-6574201800020002.

Calheiros, D. dos S., Neto, J. L. C., Melo, F. A. P. de, Pedrosa de Melo, F. Í., & Munster, M. de A. van. (2021). Quality of Life and Associated Factors Among Male Wheelchair Handball Athletes. Perceptual and Motor Skills128(4), 1623–1639. https://doi.org/10.1177/00315125211014865

Reviewer 2 Report

Manuscript ID:  ijerph 2025232

Factors associated with the lifestyle of pediatric healthcare professionals during the COVID-19 pandemic.  

Dear Authors

Many thanks for this paper. I enjoyed reading it.

The paper analyses the sociodemographic and occupational variables capable of predicting the lifestyle of health professionals in pediatric units during the COVID-19 pandemic using cross-sectional dat based on  407 health professionals

The paper is clearly relevant but lacks of sufficient motivation to justify its publication. Although as indicated there aren’t studies for other health professional, something more should be said of why we would expect health professionals in pediatrics to be affected more in their life style.

Few comments below:1

1)     Authors claim that ‘Professionals working in the field of pediatrics are exposed to a greater load of stress among other variables,’ but this does not seem to have been proved here. Also, why would that be the case?

2)     Given the focus on these four cities Salvador, Lauro de Freitas and Feira de Santana in the 117 State of Bahia, Northeast Brazil, something about their characteristics should be said. For example, are those cities more deprived so that health of children is usually worse?  I am aware estimates are controlled for a variety of variables but not at local/city level.

3)     Table 1 and 2 are confusing and unclear in the presentation, since in many variables there is no confidence intervals.

Author Response

#Reviewer 2

Dear Authors

Many thanks for this paper. I enjoyed reading it.

Thank you for the compliment.

Authors claim that ‘Professionals working in the field of pediatrics are exposed to a greater load of stress among other variables,’ but this does not seem to have been proved here. Also, why would that be the case?

We appreciate the reviewer for this insightful comment. We improved the rationale as follows:

In addition, we still do not know if there are different challenges among the pediatric team and other health work categories since the literature has got focused on the overall health workers.

Thus, the objective of this study was to identify the sociodemographic and occupational variables capable of predicting the lifestyle of health professionals in pediatric units during the COVID-19 pandemic. We believe we might build a rationale in the pediatric literature through our results. (Introduction Section, Page 3, Lines 108-114).

Given the focus on these four cities Salvador, Lauro de Freitas and Feira de Santana in the 117 State of Bahia, Northeast Brazil, something about their characteristics should be said. For example, are those cities more deprived so that health of children is usually worse?  I am aware estimates are controlled for a variety of variables but not at local/city level.

The cities are where the hospitals are placed and we did not consider the probability of taking into account the deprivation of the children who are attending in the hospitals. However, Salvador is the capital of the State and the others are nearby. The hospitals are reference in pediatric assistance in the all state and even children from small cities in the countryside are attending in these hospitals with the municipalities’ authorities and public health system supports. Unfortunately, this specific information is not part of our objectives in this study, since the professionals assist all children regardless where they are from.

Table 1 and 2 are confusing and unclear in the presentation, since in many variables there is no confidence intervals.

Dear reviewer, both tables are presenting the analysis of associations between sociodemographic variables and lifestyle (Table 1) and work variables and lifestyle (Table 2). Following your observation, we reformatted the lines and columns in order to present both tables clearly.

Reviewer 3 Report

Quality of life is one of the most popular topics of many research papers. It is usually associated with factors such as: education, economic, professional and family situation, physical fitness or sexual behavior.

So far, a huge number of standardized questionnaires examining the quality of life have been created, especially taking into account the state of health.

I congratulate the authors on the idea for the conducted research.

However, I think that the number of authors is too large. The research covers only 407 people and only one psychometric tool was used.

The relationship between the quality of life and professional work, earnings or family situation is nothing new. This has already been extensively described in the literature. 

I believe that the authors should use other tools, such as a personality test.

It is worth considering using an additional tool to study the quality of life directly related to health, e.g. The MOS 36-Item Short-Form Health Survey (SF-36).

I recommend authors to approach this particular subject of research differently (adding new research tools or trying to find other variables that can be used in statistical analysis) or finding new areas of interest. In my opinion, this article in this shape requires thorough revision to meet high quality requirements typical of IJERPH.

Author Response

#Reviewer 3

Quality of life is one of the most popular topics of many research papers. It is usually associated with factors such as: education, economic, professional and family situation, physical fitness or sexual behavior. So far, a huge number of standardized questionnaires examining the quality of life have been created, especially taking into account the state of health. I congratulate the authors on the idea for the conducted research.

We thank the reviewer for the compliment, but we want to highlight here that quality of life is different than lifestyle. Even they are related, their constructions and rationales for assessment instruments are not similar.

In order to clarify this since, the beginning of the manuscript we added the information as follows:

Lifestyle is the set of habits and customs adopted by people and communities according to their historical-cultural experiences [1], which is related to it but different from the quality of life. The World Health Organization (WHO) characterized the term quality of life as the understanding of people about their position in life, taking into account their lifestyles, goals or lifespan[2]. The main point that distinguishes the terms is the fact that quality of life is regarding people’s perceptions or conceptions about life[3], while lifestyle is regarding people’s behaviors or habits in daily life[4].  (Introduction Section, Page 2, Lines 56-62).

However, I think that the number of authors is too large. The research covers only 407 people and only one psychometric tool was used.

We understand the reviewer concerns and we would like to explain the reasons and the roles of the all 27 authors in this manuscript. With all respect, 407 people are not a small number. They are health workers from pediatric hospitals with different schedules. We needed a big team to recruit them in each hospital/place, explain them how was the project and work together with the hospital administration group as well as the group from the lab.

This study originated from a Master’s thesis of MOC, who was supervised by JLCN. The professors LHRS and MCM were on the board in this defense and provided great insightful comments on the work. DVS and CCFL are both Master’s students and helped MOC in her data collection and analysis. After the thesis defense the statistical analysis used in the manuscript was checked by DDAP, TTGD, RFFM and MERJ, who made great improvements. KAS, LELC, RASS, DSC, RGS and MCS provided a huge literature review to build along with the other authors a strong rationale of this study. LLSM, LSC, CSB and AICS worked at the hospitals and supported all the recruitment of the participants and data collection with MOC. Finally the students LSO, MMN, VABM, VFR, RAO, RDS and USL are member of the lab and helped with the data collection, data tabulation, statistical software setting and resources. They double-checked the data entry in the systems before send it to the other members of the team to analyse the data.

The relationship between the quality of life and professional work, earnings or family situation is nothing new. This has already been extensively described in the literature. 

We encourage the reviewer to take a look at the response already made in the first comment pointed out by you.

I believe that the authors should use other tools, such as a personality test.

We appreciate the reviewer for this great suggestion, but for this specific work it is not possible anymore. The data collection was made in 2021 and the current situation of the pandemic has completely changed. Maybe for further investigation in the field.

It is worth considering using an additional tool to study the quality of life directly related to health, e.g. The MOS 36-Item Short-Form Health Survey (SF-36).

Dear reviewer, this work is not about quality of life and a detailed explanation was already made in this letter. We appreciate the reviewer for this great suggestion, but for this specific work it is not possible anymore. The data collection was made in 2021 and the current situation of the pandemic has completely changed. Maybe for a further investigation in the field, which sounds really amazing for us.

I recommend authors to approach this particular subject of research differently (adding new research tools or trying to find other variables that can be used in statistical analysis) or finding new areas of interest. In my opinion, this article in this shape requires thorough revision to meet high quality requirements typical of IJERPH.

We understand the concerns pointed out by the reviewer, but we noticed merit in our work as well. Important improvements were made over the manuscript following the reviewers’ suggestions and comments. With all respect, we think the reviewer did not understand the difference between quality of life and lifestyle and that is why he/she is not seeing high quality merit in our work. We also found other similar studies of this topic published in IJERPH and why ours are not meeting the requirements?

Please take a look as follows:

Machul, M.; Bieniak, M.; ChaÅ‚daÅ›-MajdaÅ„ska, J.; BÄ…k, J.; Chrzan-Rodak, A.; Mazurek, P.; PawÅ‚owski, P.; Makuch-KuÅ›mierz, D.; Obuchowska, A.; Bartoszek, A.; Karska, K.; Jurek, K.; Cardenas, C.; Dobrowolska, B. Lifestyle Practices, Satisfaction with Life and the Level of Perceived Stress of Polish and Foreign Medical Students Studying in Poland. Int. J. Environ. Res. Public Health 202017, 4445. https://doi.org/10.3390/ijerph17124445

Descarpentrie, A.; Estevez, M.; Brabant, G.; Vandentorren, S.; Lioret, S. Lifestyle Patterns of Children Experiencing Homelessness: Family Socio-Ecological Correlates and Links with Physical and Mental Health. Int. J. Environ. Res. Public Health 202219, 16276. https://doi.org/10.3390/ijerph192316276

Bravo-Cucci, S.; Chipia, J.; Lobo, S.; López, L.; Munarriz-Medina, R.; Alvarado-Santiago, T.; Núñez-Cortés, R. Lifestyles during the First Wave of COVID-19: A Cross-Sectional Study of 16,811 Adults from Spanish-Speaking Countries in South America. Int. J. Environ. Res. Public Health 202219, 15318. https://doi.org/10.3390/ijerph192215318
